# Voxel Mamba: Group-Free State Space Models for Point Cloud based 3D Object Detection

**Guowen Zhang**[1,2], **Lue Fan**[3], **Chenhang He**[1], **Zhen Lei**[2,3,4],
**Zhaoxiang Zhang**[2,3,4,*], **Lei Zhang**[1,*]
[1]The Hong Kong Polytechnic University
[2]Centre for Artificial Intelligence and Robotics, HKISI, CAS
[3]Institute of Automation, Chinese Academy of Sciences
[4]School of Artificial Intelligence, University of Chinese Academy of Sciences
guowen.zhang@connect.polyu.hk, {csche, cslzhang}@comp.polyu.edu.hk
{lue.fan, zlei, zhaoxiang.zhang}@ia.ac.cn

## Abstract

Serialization-based methods, which serialize the 3D voxels and group them into multiple sequences before inputting to Transformers, have demonstrated their effectiveness in 3D object detection. However, serializing 3D voxels into 1D sequences will inevitably sacrifice the voxel spatial proximity. Such an issue is hard to be addressed by enlarging the group size with existing serialization-based methods due to the quadratic complexity of Transformers with feature sizes. Inspired by the recent advances of state space models (SSMs), we present a Voxel SSM, termed as Voxel Mamba, which employs a group-free strategy to serialize the whole space of voxels into a single sequence. The linear complexity of SSMs encourages our group-free design, alleviating the loss of spatial proximity of voxels. To further enhance the spatial proximity, we propose a Dual-scale SSM Block to establish a hierarchical structure, enabling a larger receptive field in the 1D serialization curve, as well as more complete local regions in 3D space. Moreover, we implicitly apply window partition under the group-free framework by positional encoding, which further enhances spatial proximity by encoding voxel positional information. Our experiments on Waymo Open Dataset and nuScenes dataset show that Voxel Mamba not only achieves higher accuracy than state-of-the-art methods, but also demonstrates significant advantages in computational efficiency. The source code is available at https://github.com/gwenzhang/Voxel-Mamba.

## 1 Introduction

LiDAR-based 3D object detection from point clouds plays an important role in applications of autonomous driving [19, 4], virtual reality [47], and robots [45]. The sparsely, unevenly and irregularly distributed point cloud data make the efficient and effective 3D object detection a very challenging task. To address these long-standing challenges, researchers have recently proposed several strategies to improve the model architecture. One strategy is to switch from PointNet-based models [49, 79, 56] to sparse convolutional neural network (SpCNN)-based models [70, 54, 10, 14, 55, 27] in order for more effective feature extraction. However, the sparse convolution is unfriendly for deployment and optimization, requiring tremendous engineering efforts. Therefore, another strategy is to switch from SpCNN to serialization-based Transformers to address this issue [13, 65, 38, 69, 42]. These methods usually group non-empty 3D voxels into multiple short sequences by serialization techniques such as

---

*Corresponding Authors.

38th Conference on Neural Information Processing Systems (NeurIPS 2024).

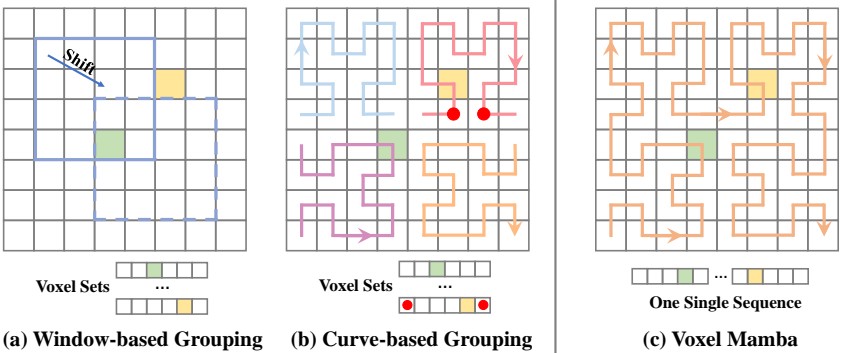

Figure 1: Comparison between (a) window-based grouping, (b) curve-based grouping, and (c) our proposed single group modeling by Voxel Mamba.

window partition [65, 13, 38], Z-shape sorting [66], and Hilbert sorting [69], as shown in Figs. 1 (a) and (b), where a sequence is a group of voxels to be processed by Transformer layers.

However, the serialization of voxels will inevitably sacrifice their spatial proximity. Some neighboring voxels can be far apart from each other after serialization, as illustrated by the two red points in Fig. 1 (b). Such a loss of proximity is difficult to be addressed in the existing serialization methods [69, 66, 38, 65, 13] because the group size is limited by the quadratic complexity of Transformers. This issue becomes even worse when neighboring voxels are grouped into different groups. Inspired by the recent success of State Space Models (SSMs) [21, 20, 59, 18, 82, 36] in language and vision, in this work we propose a simple yet effective group-free design to address the loss of proximity. Specifically, we introduce a Voxel SSM, termed as *Voxel Mamba*, for 3D object detection from point cloud. The linear computational complexity of SSMs makes it feasible to treat all voxels as a single group and sort them into a single sequence. This results in a group-free modeling of voxels, which is more efficient and deployment-friendly than previous methods since no padding tokens are needed. Nonetheless, even we can sort all voxels into a group-free sequence, it cannot be ensured that all of them are within an effective receptive field.

To enhance the spatial proximity of Voxel Mamba, we further propose two modules with it. The first is the **D**ual-scale **S**SM **B**lock (DSB) by introducing the downsampling operations in SSMs. In specific, the forward SSM branches process the high-resolution voxel features, while the backward branches extract features from the low-resolution representation. In this way, we integrate the hierarchical design with the bidirectional design in a more economical way. More importantly, the hierarchy brings a larger effective receptive field for the serialized sequence so that the spatial proximity in local 3D regions can be enhanced. The second module we introduced is the Implicit Window Partition (IWP). The window partition is a widely used strategy in previous methods [13, 65] to enhance the proximity of voxels inside a window. However, it impedes the proximity of voxels across windows and contradicts with our group-free principle. We therefore propose an implicit window partition scheme to embrace its strengths while discarding its weaknesses. In specific, we encode the voxel positions inside and across windows into embeddings for feature learning without explicitly conducting spatial window partition. In this way, better voxel proximity can be achieved under our group-free design with minimal computational cost.

Our contributions are summarized as follows:

- We propose Voxel Mamba, a group-free backbone for voxel-based 3D detection. Voxel Mamba abandons the grouping operation and serializes voxels into one single sequence, enabling better efficiency.

- To mitigate the loss of spatial proximity due to serialization, we propose the Dual-scale SSM Block (DSB) and the Implicit Window Partition (IWP) to enhance the spatial proximity preservation of Voxel Mamba.

- Our method achieves superior performance to previous state-of-the-art methods on the large-scale Waymo Open dataset [60] and nuScenes [2] datasets.

## 2    Related Work

**3D Object Detection from Point Clouds.** There are two major point cloud representations for 3D object detection, *i.e.*, point-based and voxel-based ones. As in PointNet [50, 51], point-based methods [49, 48, 56, 52, 37] directly extract geometric features from small regions of raw points. However, those methods suffer from low inference efficiency and limited context features. Voxel-based methods [54, 76, 70, 27, 79, 15–17] convert raw points into regular grids through voxelization and then process them with sparse convolution [70] or Transformers [13, 25, 65]. Voxel-based methods are currently the main stream for 3D object detection. In terms of model architecture, voxel-based methods can be categorized into two groups, *i.e.*, SpCNN-based [70, 79, 54, 55, 8, 10] and Transformers-based [65, 13, 25, 42, 38, 26] ones. Limited by the high computation complexity, SpCNN-based methods can only use small convolution kernels with restricted receptive fields, and Transformer based methods can only employ a small number of voxels in each group. In contrast, our proposed Voxel Mamba can capture long-range dependencies within the entire sequence while achieving faster inference speed than existing state-of-the-art methods.

**State Space Models.** Inspired by the continuous state space models (SSMs) in control systems, researchers [18, 21, 59, 20] have introduced the SSMs into deep neural networks as a novel alternative to CNNs and Transformers. LSSLs [22] adopts a simple sequence-to-sequence transformation, demonstrating the potential of SSMs. S4 [21] introduces a new parameterization method to SSMs to reduce the computation and memory cost. S5 [59] employs MIMO SSMs and perform efficient parallel scans based on S4. More recently, Mamba [20] introduces input-dependent SSMs and builds a generic backbone, which is fairly competitive with the well-tuned Transformers. Vision Mamba [82] employs bidirectional SSMs and position embedding to learn global visual context for vision tasks. Vmamba [36] employs a 2D-selective-scan to bridge the gap between 1D scanning and 2D plain traversing. PointMamba [34] is a pioneering work to leverage SSM for point cloud analysis, achieving impressive performance in point cloud object understanding. Subsequently, many SSM-based methods [24, 35, 78, 77, 68] are introduced for point cloud processing. In this paper, we investigate the utilization of SSMs to establish a straightforward yet robust baseline for LiDAR-based 3D object detection in driving scenes.

**Space-filling Curve.** The space-filling curve [43] is a series of fractal curves that can go through each point in a multi-dimensional space without repetition. The classical space-filling curve includes Hilbert curve [29], Z-order curve [46], and sweep curve, *etc*. Those methods can perform dimension reduction while maintaining spatial topology and locality. Many researchers [5, 69, 66, 38, 65, 3, 40] have introduced space-filling curves for point cloud processing. HilbertNet [5] uses the Hilbert curve to collapse 3D structures into 2D space to reduce computation and GPU occupation. PointGPT [3] utilizes the Morton-order curve [44] to introduce sequential properties. OctFormer [66] preserves Z-order during octreelization and adopts octree-attention for efficient context learning. PTV3 [69] streamlines the complex interaction with the space-filling curve serialization. For 3D object detection, some methods [65, 38] employ window sweep curves to group voxel features for parallel computation. We employ the Hilbert curve due to its advantageous characteristic of locality preservation.

**Point Cloud Grouping.** LiDAR point clouds are sparsely and non-uniformly distributed with varying densities. Therefore, existing methods group points or voxels to facilitate parallel computation and reduce complexity. In point cloud analysis, some works [51, 67] use the $K$ nearest neighbor (KNN) method to create groups of query points. However, the heavy computation burden makes KNN hard to scale for outdoor scenes. For 3D object detection, VoTr [42] uses a GPU-based hash table to search neighborhoods and generate fixed-length voxel groups. Window-based Voxel Transformers [13, 61, 65, 38] group voxels by employing a window-based sorting strategy, such as the rotating partition. To reduce the reliance on relative position in grouping operations, some recent works [66, 69, 40] have been proposed to group voxels based on space-filling curves. However, grouping is merely a compromise for computational complexity, which restricts the flow of information and effective receptive field. To tackle this problem, we model the entire voxels into one single sequence and allow each voxel be aware of global context information.

## 3    Methods

In this section, we present Voxel Mamba, a group-free Voxel State Space Model-based 3D backbone that can be applied to most voxel-based 3D detectors. We first introduce the preliminary concepts

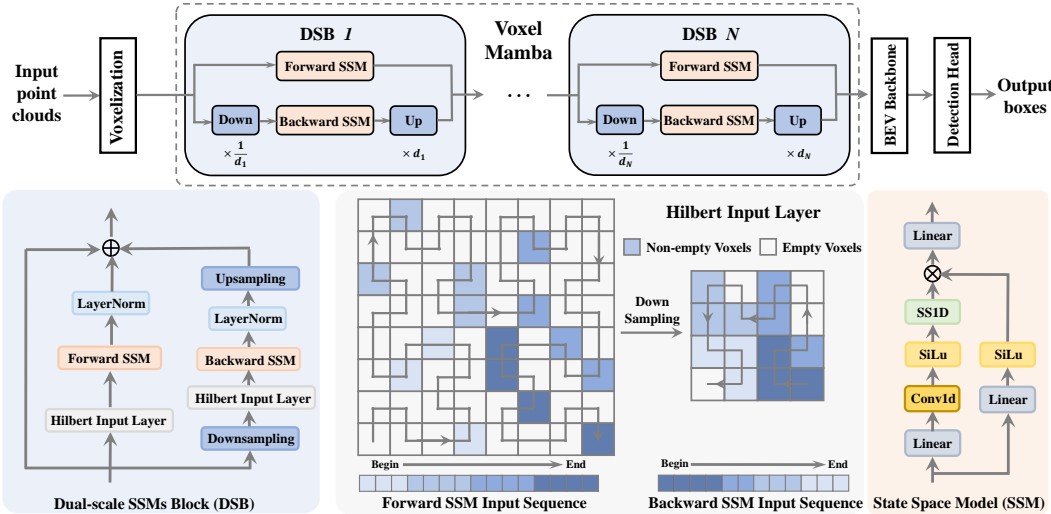

Figure 2: **Top:** The overall architecture of our proposed Voxel Mamba with $N$ Dual-scale SSM Blocks (DSBs). **Bottom:** Illustration of the DSB, including a residual connection, a forward SSM branch, and a backward SSM branch.

associated with our method, followed by the overall architecture of Voxel Mamba. Then, we describe in detail the fundamental components of Voxel Mamba, including the Hilbert Input Layer (HIL), Dual-scale SSM Block (DSB), and Implicit Window Partition (IWP).

## 3.1 Preliminaries

The state space sequence (SSM) model is a continuous-time latent state model, which maps a 1D input signal $x(t) \in \mathbb{R}^L$ to an output signal $y(t) \in \mathbb{R}^L$ through hidden state $h(t) \in \mathbb{R}^N$. The system can be represented as the following linear ordinary differential equation:

$$\begin{cases} h'(t) = \mathbf{A}h(t) + \mathbf{B}x(t), \\ y(t) = \mathbf{C}h(t) + \mathbf{D}x(t), \end{cases} \tag{1}$$

where $\mathbf{A} \in \mathbb{R}^{N \times N}$, $\mathbf{B} \in \mathbb{R}^{N \times 1}$ and $\mathbf{C} \in \mathbb{R}^{1 \times N}$ are learnable parameters, and $\mathbf{D} \in \mathbb{R}^1$ denotes a residual connection.

To apply SSM to a discrete sequence, we can discrete the continuous-time SSM with a timescale parameter $\mathbf{\Delta}$ [21, 20, 36]. The zero-order hold (ZOH) transformation can be used to discrete the continuous parameters $\mathbf{A}, \mathbf{B}$ as $\overline{\mathbf{A}} = \exp(\mathbf{\Delta A}), \overline{\mathbf{B}} = (\mathbf{\Delta A})^{-1}(\exp(\mathbf{\Delta A}) - \mathbf{I}) \cdot \mathbf{\Delta B}$. The discretized version of Eq.(1) can be written in the following recurrent form:

$$\begin{cases} h_k = \overline{\mathbf{A}}h_{k-1} + \overline{\mathbf{B}}x_k, \\ y_k = \overline{\mathbf{C}}h_k + \overline{\mathbf{D}}x_k. \end{cases} \tag{2}$$

Finally, the convolutional mode can be used for efficient parallel training:

$$\begin{cases} \overline{\mathbf{K}} = (\mathbf{C}\overline{\mathbf{B}}, \mathbf{C}\overline{\mathbf{A}}\overline{\mathbf{B}}, ..., \mathbf{C}\overline{\mathbf{A}}^{\mathbf{L}}\overline{\mathbf{B}}), \\ \mathbf{y} = \mathbf{x} * \overline{\mathbf{K}}, \end{cases} \tag{3}$$

where L is the length of the input sequence and $\overline{K} \in \mathbb{R}^L$ is the structured convolution kernel.

SSM combines the advantages of convolution and self-attention with near-linear computation and dynamic weights. It demonstrates stronger ability than Transformers in modeling long-range dependencies [21, 20], which inspires us to develop a group-free framework for point cloud based 3D object detection.

## 3.2 Overall Architecture

An overview of our proposed Voxel Mamba is shown in Figure 2. As in previous works [65, 74, 31], Voxel Mamba transforms point clouds into sparse voxels by a voxel feature encoding strategy. Unlike prior Transformer-based methods that perform extensive window partitioning and voxel grouping, in Voxel Mamba we serialize the voxel of the entire scene into a single sequence by using the *Hilbert Input Layer* (Sec. 3.3). Then, a *Dual-scale SSM Block* (Sec. 3.4) working on the voxel sequence is proposed, which allows voxels to be processed with a global context. To enlarge the effective receptive fields, DSB adopts a finer-grained perception of the voxel sequence in the forward path, and down-samples the voxels sequence in the backward path. The backward path extracts features from the low-resolution BEV representation, with an increased downsampling factor in deeper blocks. To enhance the spatial proximity in sequences, Voxel Mamba adopts *Implicit Window Partition* (Sec. 3.5) to preserve 3D positional information in the extracted voxel features, and projects them to a BEV feature map. Our proposed architecture is flexible and can be applied to most existing 3D object detection frameworks.

## 3.3 Hilbert Input Layer

The space-filling curve (*e.g.,* Hilbert [29] and Z-order [46]), known for preserving spatial locality, is widely used for dimensionality reduction. Space-filling curves, such as the Hilbert shown in Fig. 2, can traverse all elements in a space without repetition and preserve spatial topology. To improve the voxel proximity in serialization, we propose the *Hilbert Input Layer* to reorder the voxel sequence.

Denote the coordinates of voxel features as $\mathcal{C} = \{(x, y, z) \in \mathbb{R}^3 | 0 \leq x, y, z \leq n\}$. We map a voxel onto its traversal position $h$ within the Hilbert curve. Specifically, we transform $(x, y, z)$ into its binary format with $log_2 n$ bits. For example, $x$ is converted to $(x_m x_{m-1}...x_0)$, where $m = \lfloor log_2 n \rfloor$. Then, following [58], we iterate from $x_m, y_m, z_m$ to $x_1, y_1, z_1$ bits and perform exchanges and inversions to adjust the order of bits. An exchange is conducted when the current bit is 0; otherwise, an invert is conducted. We concatenate all bits as $(x_m y_m z_m x_{m-1} y_{m-1} z_{m-1} \ldots x_0 y_0 z_0)$ and apply a global $3m$-fold Gray decoding [58] on it to obtain the traversal position $h$. Subsequently, all voxels are sorted into a single sequence based on their traversal position $h$.

In our implementation, we record the traversal position $h$ corresponding to the coordinates of all potential voxels. The voxels are serialized by querying and sorting their traversal positions. We employ a distinct traversal order for each BEV resolution in Dual-scale SSM blocks. Notably, the serialization process only takes approximately 0.7ms for a sequence of length $10^6$.

## 3.4 Dual-scale SSM Block

Though space-filling curves can preserve the 3D structure to a certain degree, proximity loss is inevitable due to the dimension collapse from 3D to 1D. As a result, a local snippet of the curve can only cover a partial region in 3D space. As discussed in Sec. 1, placing all voxels in a single group cannot ensure that the effective receptive field (ERF) [41, 12] could cover all voxels. Therefore, in this subsection we introduce the Dual-scale SSM block (DSB) to build a hierarchy of state space structures and consequently improve the ERF of the model.

As shown in Fig. 2, the DSB block is designed with a residual connection [28], a forward SSM branch and a backward SSM branch. It operates on two serialized voxel sequences generated by the Hilbert Input Layer, enabling a seamless flow of information throughout the voxel sequence. The forward branch processes the original voxel sequence, maintaining high-resolution details. The backward branch, however, operates on a down-sampled voxel sequence derived from a low-resolution BEV representation. This dual-scale path allows DSB to incorporate larger-scale voxel features, enhancing the model's ability to model long dependencies among voxels. Specifically, given a voxel sequence $\mathcal{F}$ and its corresponding coordinates $\mathcal{C}$, DSB is computed as:

$$\mathcal{F}_f = \mathbf{LN}(\mathbf{FSSM}(\mathbf{HIL}(\mathcal{F} + \mathbf{IWE}(\mathcal{C})))),$$
$$\mathcal{F}_b = \mathbf{Up}(\mathbf{LN}(\mathbf{BSSM}(\mathbf{HIL}(\mathbf{Down}(\mathcal{F}) + \mathbf{IWE}(\mathcal{C}'))))), \quad (4)$$
$$\widetilde{\mathcal{F}} = \mathcal{F}_f + \mathcal{F}_b + \mathcal{F},$$

where $\mathbf{HIL}(\cdot)$ represents the Hilbert Input Layer, $\mathbf{FSSM}(\cdot)$ and $\mathbf{BSSM}(\cdot)$ denote the forward and backward SSM, $\mathbf{LN}(\cdot)$ stands for Layer Normalization, and $\mathcal{C}'$ is the coordinates of downsampled

sparse voxels. Besides, $\mathbf{Down}(\cdot)$ and $\mathbf{Up}(\cdot)$ refer to the downsampling and upsampling operations, respectively, and $\mathbf{IWE}(\cdot)$ means Implicit Window Embedding. Overall, DSB integrates the widely adopted bidirectional design [82, 36] with the hierarchical design, building sufficient receptive field to mitigate the loss of proximity without introducing additional parameters.

### 3.5 Implicit Window Partition

The window partition strategy is widely used in previous 3D detectors [13, 65] to enhance the voxel proximity. In these methods, the whole field is partitioned into multiple local windows and the voxels within a window form a group. Therefore, the voxels inside a window will have sufficient proximity; however, the voxels in different windows will have minimal proximity. In this section, we aim to introduce the advantages of window partition into our framework while avoiding its weaknesses.

To fulfill our goal, we propose an Implicit Window Partition (IWP) strategy. Unlike previous methods, we do not explicitly partition voxels into windows and apply Transformer or SSM within each window. In contrast, we calculate the voxel coordinates inside and across windows, and then encode coordinates to embeddings, termed as Implicit Window Embedding (IWE), which is formulated as:

$$\mathbf{IWE} = \mathbf{MLP}(\text{concat}(z, \lfloor \frac{x^i}{w} \rfloor, \lfloor \frac{y^i}{h} \rfloor, x^i \bmod w, y^i \bmod h)), i = 0, 1 \tag{5}$$

where $\lfloor \cdot \rfloor$ is the floor function, $w, h$ define the window shape, and $z, x^i, y^i$ are the coordinates of tokens. $(x^0, y^0)$ and $(x^1, y^1)$ represent the coordinates before and after an implicit window shift. The IWE is shared across all layers with the same stride. Thus, its computation cost only comes from shallow MLPs. With IWE, voxels in the serialized 1D curve are aware of their positions and consequently their proximity in 3D space.

### 3.6 The Voxel Mamba Backbone

With the proposed Hilbert Input Layer, DSB and IWP strategies, we build Voxel Mamba, a group-free sparse voxel backbone. The architecture of Voxel Mamba is illustrated in Figure 2. It comprises $N$ DSB blocks, which are organized into different stages based on their downsampling rates. SpConv [9] is employed to progressively decrease the feature map resolution along the Z-axis in each stage. Before sparse tokens are fed into the BEV backbone, we scatter them into dense BEV features. On the Waymo dataset, we adopt the BEV backbone from Centerpoint-Pillar [74], and employ the same setting as DSVT [65] for the detection head and loss function. On the nuScenes dataset, we only replace the 3D backbone of DSVT [65] with our Voxel Mamba backbone.

## 4 Experiments

### 4.1 Datasets and Evaluation Metrics

**Waymo Open Dataset** contains 230k annotated samples, partitioned into 160k for training, 40k for validation and 30k for testing. Each frame covers a large perception range ($150m \times 150m$). The mean average precision (mAP) and its weighted variant by heading accuracy (mAPH) are used as evaluation metrics. They are further categorized into Level 1 for objects detected by over five points, and Level 2 for those detected with at least one point.

**nuScenes** consists of 40k labeled samples, with 28k for training, 6k for validation and 6k for testing. For 3D object detection, nuScenes employs the mean average precision (mAP) and the nuScenes detection score (NDS) to measure model performance.

### 4.2 Implementation Details

Our method is implemented based on the open-source framework OpenPCDet [63]. The voxel sizes are defined as $(0.32m, 0.32m, 0.1875m)$ for Waymo and $(0.3m, 0.3m, 0.2m)$ for nuScenes. We stack six DSB blocks, divided into three stages, for the Voxel Mamba backbone network. The downsampling rates for DSBs' backward branches in each stage are $\{1, 2, 4\}$. Specifically, we employ SpConv [9] and its counterpart SpInverseConv as downsampling and upsampling operators in the DSB backward branch. On the Waymo dataset, we follow the training schemes in [65, 74] to optimize

Table 1: Performance comparison on the **validation** set of Waymo Open Dataset (single-frame setting). Symbol '-' means that the result is not available.

| Method | Category | ALL (3D mAPH) | | Vehicle (AP/APH) | | Pedestrian (AP/APH) | | Cyclist (AP/APH) | |
|---|---|---|---|---|---|---|---|---|---|
| | | L1 | L2 | L1 | L2 | L1 | L2 | L1 | L2 |
| PointPillar [31] | | 63.3 | 57.5 | 71.6 / 71.0 | 63.1 / 62.5 | 70.6 / 56.7 | 62.9 / 50.2 | 64.4 / 62.3 | 61.9 / 59.9 |
| Centerpoint-Pillar [74] | 2D CNN | - | - | 76.1 / 75.5 | 68.0 / 67.5 | - | - / 62.6 | - | - / 67.6 |
| PillarNeXt [32] | | 75.7 | 69.7 | 78.4 / 77.9 | 70.3 / 69.8 | 82.5 / 77.1 | 74.9 / 69.8 | 73.2 / 72.2 | 70.6 / 69.6 |
| SECOND [70] | | 63.1 | 57.2 | 72.3 / 71.7 | 63.9 / 63.3 | 68.7 / 58.2 | 60.7 / 51.3 | 60.6 / 59.3 | 58.3 / 57.1 |
| Part-A2 [57] | | 70.3 | 63.8 | 77.1 / 76.5 | 68.5 / 68.0 | 75.2 / 66.9 | 66.2 / 58.6 | 68.6 / 67.4 | 66.1 / 64.9 |
| PV-RCNN [54] | | 69.6 | 63.3 | 77.5 / 76.9 | 69.0 / 68.4 | 75.0 / 65.7 | 66.0 / 57.6 | 67.8 / 66.4 | 65.4 / 64.0 |
| Centerpoint-Voxel [74] | | - | 67.6 | 76.6 / 76.0 | 68.9 / 68.4 | 79.0 / 73.4 | 71.0 / 65.8 | 72.1 / 71.0 | 69.5 / 68.5 |
| PV-RCNN++ [55] | | 75.2 | 68.6 | 79.1 / 78.6 | 70.3 / 69.9 | 80.6 / 74.6 | 71.9 / 66.3 | 73.5 / 72.4 | 70.7 / 69.6 |
| AFDetV2[30] | SpCNN | 74.8 | 68.8 | 77.6 / 77.1 | 69.7 / 69.2 | 80.2 / 74.6 | 72.2 / 67.0 | 73.7 / 72.7 | 71.0 / 70.1 |
| VoxelNeXt [8] | | 76.3 | 70.1 | 78.2 / 77.7 | 69.9 / 69.4 | 81.5 / 76.3 | 73.5 / 68.6 | 76.1 / 74.9 | 73.3 / 72.2 |
| HEDNet [75] | | 79.4 | 73.4 | 81.1 / 80.6 | 73.2 / 72.7 | 84.4 / 80.0 | 76.8 / 72.6 | 78.7 / 77.7 | 75.8 / 74.9 |
| PillarNet [53] | | 74.6 | 68.4 | 79.1 / 78.6 | 70.9 / 70.5 | 80.6 / 74.0 | 72.3 / 66.2 | 72.3 / 71.2 | 69.7 / 68.7 |
| FSD [14] | | 77.3 | 70.8 | 79.2 / 78.8 | 70.5 / 70.1 | 82.6 / 77.3 | 73.9 / 69.1 | 77.1 / 76.0 | 74.4 / 73.3 |
| ConQueR [81] | | 77.9 | 71.6 | 78.4 / 77.9 | 71.0 / 70.5 | 82.4 / 76.6 | 75.8 / 70.1 | 77.5 / 76.4 | 75.2 / 74.1 |
| VoTR [42] | | - | - | 75.0 / 74.3 | 65.9 / 65.3 | - | - | - | - |
| VoxSeT [25] | | 72.2 | 66.2 | 74.5 / 74.0 | 66.0 / 65.6 | 80.0 / 72.4 | 72.5 / 65.4 | 71.6 / 70.3 | 69.0 / 67.7 |
| SST[13] | | - | - | 76.2 / 75.8 | 68.0 / 67.6 | 81.4 / 74.1 | 72.8 / 65.9 | - | - |
| SWFormer[61] | | - | - | 77.8 / 77.3 | 69.2 / 68.8 | 80.9 / 72.7 | 72.5 / 64.9 | - | - |
| CenterFormer [80] | Group-based | 73.2 | 69.1 | 75.0 / 74.4 | 69.9 / 69.4 | 78.0 / 72.4 | 73.1 / 67.7 | 73.8 / 72.7 | 71.3 / 70.2 |
| FlatFormer [38] | | - | 67.2 | - | 69.0 / 68.6 | - | 71.5 / 65.3 | - | 68.6 / 67.5 |
| PTv3 [69] | | - | 70.5 | - | 71.2 / 70.8 | - | 76.3 / 70.4 | - | 71.5 / 70.4 |
| DSVT-Voxel [65] | | 78.2 | 72.1 | 79.7 / 79.3 | 71.4 / 71.0 | 83.7 / 78.9 | 76.1 / 71.5 | 77.5 / 76.5 | 74.6 / 73.7 |
| **Voxel Mamba** (ours) | Group-free | **79.6** | **73.6** | 80.8 / 80.3 | 72.6 / 72.2 | 85.0 / 80.8 | 77.7 / 73.6 | 78.6 / 77.6 | 75.7 / 74.8 |

Table 2: Performance comparison on the **test** set of Waymo Open Dataset. Symbol '-' means that the result is not available. "3f" stands for 3-frame model.

| Method | ALL (3D mAPH) | | Vehicle (AP / APH) | | Pedestrian (AP / APH) | | Cyclist (AP / APH) | |
|---|---|---|---|---|---|---|---|---|
| | L1 | L2 | L1 | L2 | L1 | L2 | L1 | L2 |
| PointPillar[31] | - | - | 68.6 / 68.1 | 60.5 / 60.1 | 68.0 / 55.5 | 61.4 / 50.1 | - | - |
| M3DETR[23] | 67.1 | 61.9 | 77.7 / 77.1 | 70.5 / 70.0 | 68.2 / 58.5 | 60.6 / 52.0 | 67.3 / 65.7 | 65.3 / 63.8 |
| 3D-MAN [73] | - | - | 78.7 / 78.3 | 70.4 / 70.0 | 70.0 / 66.0 | 64.0 / 60.3 | - | - |
| PV-RCNN++ [55] | 75.7 | 70.2 | 81.6 / 81.2 | 73.9 / 73.5 | 80.4 / 75.0 | 74.1 / 69.0 | 71.9 / 70.8 | 69.3 / 68.2 |
| CenterPoint [74] | 77.2 | 71.9 | 81.1 / 80.6 | 73.4 / 73.0 | 80.5 / 77.3 | 74.6 / 71.5 | 74.6 / 73.7 | 72.2 / 71.3 |
| RSN [62] | - | - | 80.7 / 80.3 | 71.9 / 71.6 | 78.9 / 75.6 | 70.7 / 67.8 | - | - |
| SST-3f [13] | 78.3 | 72.8 | 81.0 / 80.6 | 73.1 / 72.7 | 83.3 / 79.7 | 76.9 / 73.5 | 75.7 / 74.6 | 73.2 / 72.2 |
| Graph-RCNN [71] | 77.0 | 71.6 | 83.6 / 83.1 | 76.0 / 75.6 | 81.9 / 76.5 | 75.6 / 70.5 | 72.5 / 71.3 | 69.8 / 68.7 |
| FSDv1 [14] | 78.2 | 72.4 | 82.7 / 82.3 | 74.4 / 74.1 | 82.9 / 77.9 | 75.9 / 71.3 | 75.6 / 74.4 | 72.9 / 71.8 |
| FSDv2 [15] | 79.0 | 73.3 | 82.4 / 82.0 | 74.4 / 74.0 | 83.8 / 78.9 | 77.4 / 72.8 | 77.1 / 76.0 | 74.3 / 73.2 |
| PillarNeXt-3f [32] | 79.0 | 74.1 | 83.3 / 82.8 | 76.2 / 75.8 | 84.4 / 81.4 | 78.8 / 76.0 | 73.8 / 72.7 | 71.6 / 70.6 |
| **Voxel Mamba** (ours) | **79.6** | **74.3** | 84.4 / 84.0 | 77.0 / 76.6 | 84.8 / 80.6 | 79.0 / 74.9 | 75.4 / 74.3 | 72.6 / 71.5 |

the model using Adam optimizer with weight decay 0.05, one-cycle learning rate policy, max learning rate 0.0025, and batch size 24 for 24 epochs. On the nuScenes dataset, we follow the training scheme adopted in DSVT [65]. We train Voxel Mamba with a weight decay of 0.05, one-cycle learning rate policy, max learning rate of 0.004, and batch size of 32 for 20 epochs. The voxel features on both datasets consist of 128 channels. All the models are trained on 8 RTX A6000 GPUs. Other settings in the training and inference of Voxel Mamba follow DSVT [65].

## 4.3 Comparison with State-of-the-art Methods

**Waymo.** We first compare Voxel Mamba with state-of-the-art methods on the Waymo Open dataset. Table 1 shows the results on the validation set. Our proposed Voxel Mamba achieves 79.6/73.4 on L1/L2 mAPH, which are +1.4 and +1.5 better than DSVT-Voxel. Since our framework differs from DSVT only on the 3D backbone, it can be concluded that Voxel Mamba has superior ability in capturing voxel features. In comparison with window-based (*e.g.,* DSVT) or curve-based (*e.g.,* PTv3) grouping methods, our group-free method Voxel Mamba consistently delivers better results. Table 2 shows the results on the test split. Voxel Mamba reaches 79.6/74.3 in terms of L1/L2 mAPH, which is even better than the 3-frame setting of PillarNeXt and SST.

**nuScenes.** We then compare Voxel Mamba with previous state-of-the-art methods on nuScenes. Table 3 shows the results on the validation set. Voxel Mamba achieves impressive results with 71.9

Table 3: Comparison with the state-of-the-art detectors on the nuScenes dataset **validation** split.

| Method | NDS | mAP | Car | Truck | Bus | T.L. | C.V. | Ped. | M.T. | Bike | T.C. | B.R. |
|---|---|---|---|---|---|---|---|---|---|---|---|---|
| CenterPoint [74] | 66.5 | 59.2 | 84.9 | 57.4 | 70.7 | 38.1 | 16.9 | 85.1 | 59.0 | 42.0 | 69.8 | 68.3 |
| VoxelNeXt [8] | 66.7 | 60.5 | 83.9 | 55.5 | 70.5 | 38.1 | 21.1 | 84.6 | 62.8 | 50.0 | 69.4 | 69.4 |
| TransFusion-L [1] | 70.1 | 65.5 | 86.9 | 60.8 | 73.1 | 43.4 | 25.2 | 87.5 | 72.9 | 57.3 | 77.2 | 70.3 |
| PillarNeXt [32] | 68.4 | 62.2 | 85.0 | 57.4 | 67.6 | 35.6 | 20.6 | 86.8 | 68.6 | 53.1 | 77.3 | 69.7 |
| HEDNet [75] | 71.4 | 66.7 | 87.7 | 60.6 | 77.8 | 50.7 | 28.9 | 87.1 | 74.3 | 56.8 | 76.3 | 66.9 |
| DSVT [65] | 71.1 | 66.4 | 87.4 | 62.6 | 75.9 | 42.1 | 25.3 | 88.2 | 74.8 | 58.7 | 77.8 | 70.9 |
| **Voxel Mamba** (ours) | **71.9** | **67.5** | 87.9 | 62.8 | 76.8 | 45.9 | 24.9 | 89.3 | 77.1 | 58.6 | 80.1 | 71.5 |

Table 4: Comparison with the state-of-the-art detectors on the nuScenes dataset **test** split.

| Method | NDS | mAP | Car | Truck | Bus | T.L. | C.V. | Ped. | M.T. | Bike | T.C. | B.R. |
|---|---|---|---|---|---|---|---|---|---|---|---|---|
| PointPillars [31] | 45.3 | 30.5 | 68.4 | 23.0 | 28.2 | 23.4 | 4.1 | 59.7 | 27.4 | 1.1 | 30.8 | 38.9 |
| 3DSSD [72] | 56.4 | 42.6 | 81.2 | 47.2 | 61.4 | 30.5 | 12.6 | 70.2 | 36.0 | 8.6 | 31.1 | 47.9 |
| CenterPoint [74] | 65.5 | 58.0 | 84.6 | 51.0 | 60.2 | 53.2 | 17.5 | 83.4 | 53.7 | 28.7 | 76.7 | 70.9 |
| FCOS-LiDAR [64] | 65.7 | 60.2 | 82.2 | 47.7 | 52.9 | 48.8 | 28.8 | 84.5 | 68.0 | 39.0 | 79.2 | 70.7 |
| AFDetV2 [30] | 68.5 | 62.4 | 86.3 | 54.2 | 62.5 | 58.9 | 26.7 | 85.8 | 63.8 | 34.3 | 80.1 | 71.0 |
| UVTR-L [33] | 69.7 | 63.9 | 86.3 | 52.2 | 62.8 | 59.7 | 33.7 | 84.5 | 68.8 | 41.1 | 74.7 | 74.9 |
| VISTA [11] | 69.8 | 63.0 | 84.4 | 55.1 | 63.7 | 54.2 | 25.1 | 82.8 | 70.0 | 45.4 | 78.5 | 71.4 |
| Focals Conv [6] | 70.0 | 63.8 | 86.7 | 56.3 | 67.7 | 59.5 | 23.8 | 87.5 | 64.5 | 36.3 | 81.4 | 74.1 |
| VoxelNeXt [8] | 70.0 | 64.5 | 84.6 | 53.0 | 64.7 | 55.8 | 28.7 | 85.8 | 73.2 | 45.7 | 79.0 | 74.6 |
| TransFusion-L [1] | 70.2 | 65.5 | 86.2 | 56.7 | 66.3 | 58.8 | 28.2 | 86.1 | 68.3 | 44.2 | 82.0 | 78.2 |
| LinK [39] | 71.0 | 66.3 | 86.1 | 55.7 | 65.7 | 62.1 | 30.9 | 85.8 | 73.5 | 47.5 | 80.4 | 75.5 |
| HEDNet [75] | 72.0 | 67.7 | 87.1 | 56.5 | 70.4 | 63.5 | 33.6 | 87.9 | 70.4 | 44.8 | 85.1 | 78.1 |
| LargeKernel3D [7] | 70.6 | 65.4 | 85.5 | 53.8 | 64.4 | 59.5 | 29.7 | 85.9 | 72.7 | 46.8 | 79.9 | 75.5 |
| PillarNet [53] | 71.4 | 66.0 | 87.6 | 57.5 | 63.6 | 63.1 | 27.9 | 87.3 | 70.1 | 42.3 | 83.3 | 77.2 |
| FSDv2 [15] | 71.7 | 66.2 | 83.7 | 51.6 | 66.4 | 59.1 | 32.5 | 87.1 | 71.4 | 51.7 | 80.3 | 78.7 |
| DSVT [65] | 72.7 | 68.4 | 86.8 | 58.4 | 67.3 | 63.1 | 37.1 | 88.0 | 73.0 | 47.2 | 84.9 | 78.4 |
| **Voxel Mamba** (ours) | **73.0** | **69.0** | 86.8 | 57.1 | 68.0 | 63.2 | 35.4 | 89.5 | 74.7 | 50.8 | 86.9 | 77.3 |

NDS and 67.5 mAP, which is +0.5 and +0.8 higher than the previous best method. Compared with DSVT, Voxel Mamba achieves +1.1 higher performance on mAP. The results on the test split are shown in Table 4. Our method also exhibits the best mAP and NDS.

**Inference Efficiency.** We compare Voxel Mamba with other state-of-the-art methods in inference speed and performance accuracy in Fig. 3. Notably, Voxel Mamba outperforms DSVT [65] and PV-RCNN++ [54] by at least +1.5 in detection accuracy, while achieving faster speed. Some methods, such as CenterPoint [74] and PointPillar [31], are faster than Voxel Mamba; however, their accuracy is substantially lower.

We further compare Voxel Mamba with previous well-designed architectures (SpCNN, Transformers, and 2D CNN) in GPU memory in Table 5. Compared with CenterPoint-Pillar, Voxel Mamba requires only an additional 0.5 GB GPU memory but achieves +9.0 higher accuracy in L2 mAPH. While Transformer-based methods like SST [13] and DSVT [65] use group partitioning, they still consume more memory than our group-free Voxel Mamba. All the experiments are evaluated on an NVIDIA A100 GPU with the same environment.

## 4.4 Ablation Studies

To better investigate the effectiveness of Voxel Mamba, we conduct a set of ablation studies by using the nuScenes validation set. We follow OpenPCDet [63] to train all models for 20 epochs.

**Effectiveness of Space-filling Curves.** There are some potential alternatives to Hilbert curve for preserving locality. Here, we compare Hilbert curve with some commonly used space-filling curves (Z-order [66] and window partition [13]) in 3D detection. As shown in Table 6(a), without using space-filling curves (*i.e.*, the row of 'Random Curve'), there will be a notable decline in performance, which indicates that spatial proximity is crucial in the group-free setting. By using the Z-order curve

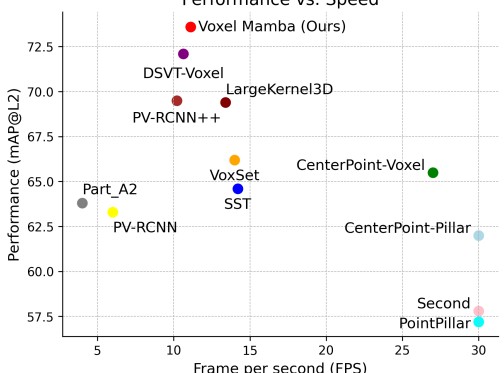

Figure 3: Detection performance (mAPH/L2) vs. speed (FPS) on Waymo.

Table 5: Comparison with other well-designed architectures on GPU memory.

| Abalation | Backbone | Memory (GB) |
|---|---|---|
| PointPillar [31] | 2D CNN | 3.6 |
| Centerpoint-Pillar [74] | | 3.2 |
| Part-A2 [57] | SpCNN | 2.9 |
| PV-RCNN++ (ResNet) [55] | | 17.2 |
| SST [13] | Transformers | 6.8 |
| DSVT-Voxel [65] | | 4.2 |
| Voxel Mamba (Ours) | SSMs | 3.7 |

Table 6: Ablations on the nuScenes validation split. In (d), Centerpoint-Pillar is used as the baseline.

| Space-filing Curve | mAP | NDS |
|---|---|---|
| Random Curve | 66.0 | 71.0 |
| Window Partition | 67.3 | 71.7 |
| Z-Order Curve | 67.0 | 71.7 |
| Hilbert Curve | **67.5** | **71.9** |

(a) Ablation on space-filling curves.

| Ablation | mAP | NDS |
|---|---|---|
| Baseline | 63.3 | 69.1 |
| + bidirectional SSMs (Hilbert curve) | 65.8 | 70.9 |
| + Voxel | 66.3 | 71.0 |
| + DSB | 66.7 | 71.3 |
| + IWP | **67.5** | **71.9** |

(b) Effect of each component in Voxel Mamba.

| Downstrides | mAP | NDS |
|---|---|---|
| {1,1,1} | 66.6 | 71.4 |
| {1,2,2} | 66.9 | 71.8 |
| {2,2,2} | 65.6 | 70.8 |
| {4,4,4} | 66.2 | 71.2 |
| {1,2,4} | **67.5** | **71.9** |

(c) Ablation on the downsampling rates of DSB.

| Pos Embeding | mAP | NDS |
|---|---|---|
| Baseline | 66.7 | 71.3 |
| Absolute position | 66.9 | 71.2 |
| Cos, Sin | 66.6 | 71.4 |
| Ours (w/o shift) | 67.3 | 71.9 |
| Ours | **67.5** | **71.9** |

(d) Ablation on IWE.

and window partition to introduce spatial proximity, the mAP and NDS are much improved. The serialization based on the Hilbert curve can further enhance the model performance.

**Effectiveness of Each Component.** To more clearly illustrate the effectiveness of the different components in Voxel Mamba, we conduct experiments by adding each of them to a baseline, which is set to Centerpoint-Pillar [74]. As shown in Table 6(b), bidirectional SSMs with a Hilbert-based group-free sequence can significantly improve the accuracy over the baseline, which validates the feasibility of our group-free strategy. Besides, converting pillar to voxel can enhance much the detector's performance without group size constraints. Voxel Mamba with DSB obtain better performance than the plain bidirectional SSMs. This is because DSB can build larger ERFs and mitigate the loss of proximity. Furthermore, IWE further boosts Voxel Mamba's performance for its capability in capturing 3D position information and increasing voxel proximity.

**Downsampling Rates of DSB.** We evaluate the impact of different downsampling rates in DSB by adjusting the stride $\{d_1, d_2, d_3\}$ in the backward SSM branch at each stage. $d_i = 1$ means the original resolution is used. The results are shown in Table 6(c). We see that transitioning from {1,1,1} to {1,2,2} and to {1,2,4} enhances performance due to an enlarged effective receptive field and improved proximity by using larger downsampling rates at late stages. However, DSBs with {2,2,2} or {4,4,4} compromise performance compared to {1,1,1}, indicating that using larger downsampling rates at early stages will lose some fine details. Thus, we set the stride as {1,2,4} to strike a balance between effective receptive fields and detail preservation.

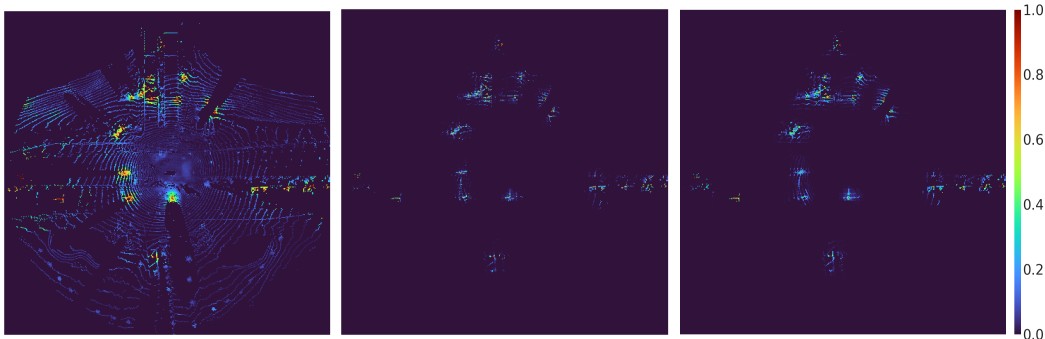

Figure 4: The effective receptive fields (ERFs) of Voxel Mamba (left), group-based bidirectional Mamba (middle) and DSVT (right).

**Effectiveness of IWE.** Table 6(d) validates the capability of IWE to enhance spatial proximity. We compare IWE with some commonly used positional embedding methods [13, 65] in 3D detection. Absolute position denotes the direct encoding of voxel coordinates using an MLP. The results demonstrate that IWE can significantly improve the detection performance by offering features with rich 3D positional and proximate information.

### 4.5 Effective Receptive Field of Voxel Mamba

Fig. 4 illustrates the *Effective Receptive Fields* [41, 12] (ERFs) of window partition-based method DSVT [65], group-based bidirectional Mamba and our proposed group-free method Voxel Mamba. For clear visualization, all models take pillars as inputs. The group partition in the group-based bidirectional Mamba is configured identically to DSVT. Then, we randomly select voxels of interest from the ground truth bounding box and calculate the ERF at each non-empty voxel position. Subsequently, we merge the ERFs into a single image by taking the maximum value at each voxel location. A wider activation area indicates a larger ERF. From Fig. 4, we see that Voxel Mamba exhibits a notably larger ERF than DSVT and group-based bidirectional Mamba, which can be attributed to the benefits of group-free operation. The larger ERF can cover a more complete local region and enhance the spatial proximity in 1D sequences.

## 5   Conclusion

In this paper, we proposed Voxel Mamba, a group-free SSM-based 3D backbone for point cloud based 3D detection. We first analyzed the proximity loss of group partition in current serialization-based 3D detection methods. By taking the advantage of linear complexity of SSMs, we proposed a group-free strategy to alleviate the loss of spatial proximity in 3D to 1D serialization. We further proposed the DSB block and IWP strategy to build larger effective receptive fields and improve the spatial proximity of our Voxel Mamba framework. Experiments demonstrated that Voxel Mamba achieved state-of-the-art results on Waymo and nuScene datasets. Without elaborated optimization, our model consumed less memory than group-based Voxel Transformer methods, and our group-free strategy was more efficient and deployment-friendly than group partition. Voxel Mamba provided an efficient group-free solution for sparse point clouds for 3D tasks.

**Limitations.** While the proposed Voxel Mamba achieves state-of-the-art performance in point cloud based 3D object detection, it still has some limitations to be further addressed. First, in the Hilbert Input Layer, the curve templates occupy approximately 0.1 GB of GPU memory, which may become substantial as the voxel resolution increases. Besides, a more elaborately designed downsampling and upsampling operation could improve more the model efficiency. We will investigate these problems in future work.

**Acknowledgments.** This work was supported in part by the InnoHK Program.

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
