# OpenReview forum: "Voxel Mamba: Group-Free State Space Models for Point Cloud based 3D Object Detection"
_NeurIPS.cc/2024/Conference — NeurIPS 2024 spotlight_

### Official Review · Reviewer_rV2B · 2024-07-03

**Soundness:** 2
**Presentation:** 2
**Contribution:** 3
**Rating:** 5
**Confidence:** 4

**Summary:**

The paper focuses on a new architecture, Voxel Mamba, for 3D object detection using point clouds. It introduces a group-free approach utilizing state space models (SSMs) to address the limitations of previous voxel-based methods like small receptive fields and inefficient grouping. Voxel Mamba leverages asymmetrical SSMs for handling multi-scale contexts and employs space-filling curves for efficient data serialization. They have evaluated the method on the Waymo and nuScenes datasets.

**Strengths:**

1. Integrating group-free state space models and space-filling curves significantly advances traditional voxel-based methods. This allows for better handling of long-range dependencies and multi-scale contexts.
2. The manuscript provides comprehensive empirical evidence, showing improvements in accuracy and efficiency on prominent benchmarks (nuScenes and Waymo).
3. The use of the Hilbert curve for input serialization and the introduction of asymmetrical state space models are well-explained and demonstrate a deep understanding of the challenges in 3D object detection.

**Weaknesses:**

1. There are typos, e.g., Line 293: "finner" should be "finer";
2. The manuscript depth, while comprehensive, might be overwhelming for readers unfamiliar with the domain, particularly the advanced concepts like Hilbert curves and asymmetrical state space models. It also might pose challenges in implementation, especially for teams with limited expertise in these areas.
3. The Voxel Mamba might require substantial computational resources, e.g., GPU memory and processing power, which could limit its deployment in constrained environments.
4. The complexity of the Voxel Mamba model doesn't yield proportionately significant performance improvements. Voxel Mamba achieves superior performance on nuScenes and Waymo. However, while statistically significant, the improvement margins might be weighed against the increase in model complexity, implementation difficulty, and computational resource requirements.

**Questions:**

1. How does the model perform under different weather conditions or with varying light levels, which are common challenges in outdoor 3D object detection?
2. What specific modifications would be necessary to adapt this model for indoor use, where point cloud data might come from different types of sensors or have different characteristics like higher density?
3. Can Voxel Mamba operate in a real-time setting given its computational demands? What are the latency metrics when deployed in such environments?

**Limitations:**

See the above "Questions" and "Weaknesses" parts.

---

> ### Author Rebuttal · Authors · 2024-08-07
>
> **[Q1, Q2] Manuscript refinement.**
>
> We apologize for the typos and presentation issues. We promise to thoroughly proofread the entire manuscript and correct the grammatical and typographical errors as much as possible. Considering that some concepts may be difficult for newcomers to this field, we will include more intuitive explanations and visualizations to aid their understanding. For sure we will release our code and pre-trained models to provide detailed implementation.
>
> **[Q3, Q4] Computational complexity.**
>
> Firstly, it's important to clarify that Voxel Mamba's computational complexity significantly differs from Mamba in language models. In our implementation, its layers and parameters are carefully aligned with standard 3D object detectors.
>
> Besides, in Table 5(e) and Figure 3 of the main paper, we compared our method with previous state-of-the-art (SOTA) methods in terms of latency, performance, and GPU memory. Here, we provide a more detailed comparison between Voxel Mamba and previous SOTA serialization-based methods. Actually, Voxel Mamba has comparable to or even lower computational demands than other SOTA serialization-based methods. As shown in the following table, Voxel Mamba differs from DSVT only in the 3D backbone but outperforms DSVT by +1.5 L2 mAPH with less GPU memory and inference latency. All the results are tested on the same device.
>
> | Ablation       | Type     | Backbone      | Memory (GB) | Latency (ms) | L2 mAPH    |
> |----------------|----------|---------------|-------------|--------------|------------|
> | SST            | Pillar   | Transformers  | 6.8         | 71           | 64.6       |
> | VoxSet         | Pillar   | Transformers  | 5.3         | 73           | 67.7       |
> | Voxel Mamba    | Pillar   | SSMs          | 3.5         | 66           | **71.2**   |
> | DSVT           | Voxel    | Transformers  | 4.2         | 94           | 72.1       |
> | Voxel Mamba    | Voxel    | SSMs          | 3.7         | 90           | **73.6**   |
>
> **[Q5] Robustness to weather and light.**
>
> Thanks for the constructive suggestion. Adverse weather can cause scattered points due to attenuation and backscattering. To evaluate Voxel Mamba's robustness to adverse weather, we simulated various fog densities using the fog augmentation method in [1] on the point clouds during inference. The table below shows the mAP results on the nuScene dataset with fog severity from 1 to 5 (heavy). One can see that Voxel Mamba consistently outperforms DSVT across all fog densities. Moreover, with the increase of severity, the performance advantage of Voxel Mamba increases, indicating our method's superior robustness to adverse weather.
>
> | Method        | Clean | Severity 1 | Severity 2 | Severity 3 | Severity 4 | Severity 5 |
> |---------------|-------|------------|------------|------------|------------|------------|
> | DSVT          | 66.4  | 65.6       | 63.4       | 55.3       | 47.2       | 25.1       |
> | Voxel Mamba   | 67.5 (+1.1) | 66.9 (+1.3) | 64.9 (+1.5) | 58.0 (+2.7) | 50.7 (+3.5) | 26.8 (+1.7) |
>
> [1] Fog Simulation on Real LiDAR Point Clouds for 3D Object Detection in Adverse Weather. ICCV21
>
> **[Q6] Indoor semantic segmentation.**
>
> Thanks for the question. Please kindly refer to **shared response Q3** for details.
>
> **[Q7] Real-time Voxel Mamba.**
>
> It is a common practice to optimize 3D detection algorithms for deployment in autonomous driving systems. Frameworks such as TensorRT are widely used to enhance the memory efficiency and inference speed. Considering that the frequency of outdoor LiDAR sensors ranges from 5 to 20 Hz, the optimized Voxel Mamba can indeed meet the real-time requirements.

---

> > ### Comment · Area_Chair_SwCv · 2024-08-11
> >
> > Dear Reviewer,
> >
> > This is a gentle reminder to please review the rebuttal provided by the authors. Your feedback is crucial to the decision-making process. Please consider updating your score after reading the rebuttal.
> >
> > Thank you for your help with the NeurIPS!
> >
> > Best, Your AC

---

### Official Review · Reviewer_vX3v · 2024-07-06

**Soundness:** 2
**Presentation:** 2
**Contribution:** 2
**Rating:** 5
**Confidence:** 5

**Summary:**

The paper introduces a novel architecture for 3D object detection, addressing the limitations of current methods which rely on grouping operations. The proposed Voxel Mamba leverages State Space Models (SSMs) to capture long-range dependencies and process the entire scene in a single sequence, avoiding inefficient grouping. It employs a group-free strategy, an Asymmetrical State Space Models (ASSMs) block for multi-scale context, and an Implicit Window-based Position Embedding (IWPE) to maintain token locality. Experiments on the Waymo Open and nuScenes datasets demonstrate superior performance in accuracy and efficiency over established methods.

**Strengths:**

1.	The Voxel Mamba's group-free strategy is innovative, potentially offering a more efficient and effective approach to handling point cloud data compared to traditional grouping methods.
2.	The use of State Space Models (SSMs) for point cloud processing is a novel concept that shows promise for capturing long-range dependencies in 3D object detection tasks.
3.	The Asymmetrical State Space Models (ASSMs) block cleverly integrates multi-scale context into the model, which likely enhances the feature representation and detection accuracy.
4.	Window-based Position Embedding (IWPE) is a thoughtful addition to maintain the fine-grained 3D position information crucial for accurate object detection.
5.	The architecture's flexibility to integrate into existing detection frameworks is a significant advantage, as it allows for the incremental adoption of Voxel Mamba in various systems.
6.	The extensive experimental evaluation on large-scale and complex datasets like Waymo Open and nuScenes is commendable and provides strong evidence of the model's capabilities.

**Weaknesses:**

1.	The introduction does not clearly explain what problem the introduction of the State Space Model is aimed at solving, the motivation of the article, and the advantages of Memba.
2.	While the paper claims high efficiency, specific computational metrics such as runtime and memory usage for different model sizes or input resolutions are not provided for a comprehensive assessment.
3.	The paper does not provide visual experimental results.
4.	The generalization of Voxel Mamba's performance to other datasets or domains beyond autonomous driving scenarios is not discussed, limiting the understanding of its versatility.
5.	The paper lacks a detailed comparison with other state-of-the-art methods in terms of computational efficiency, which is critical for applications with limited computational resources.
6.	The robustness of the model to variations in point cloud density and sparsity is not thoroughly investigated, which could be a concern for outdoor scenes with varying environmental conditions.

**Questions:**

1.	The role of using downsampling sequences in the backward SSM sequence in Figure 2 is not clearly explained in the paper.What does downsampling do?
2.	Are there any improvements made to the object detection baseline, including loss function and pipeline?
3.	Is there any redundancy in your title? "based 3D Object Detection" itself is a task of 3D point cloud, why is it written "for Point Cloud" in front of it?
4.	How does the computational complexity of Voxel Mamba scale with the size and density of the point cloud data?
5.	What are the specific advantages of using SSMs over traditional CNN or Transformer architectures in the context of 3D object detection from point clouds?
6.	Can the authors provide more details on the design choices behind the Implicit Window-based Position Embedding and its impact on the model's receptive field?
7.	How does Voxel Mamba handle dynamic objects or scenes with non-static elements, which are common in autonomous driving scenarios?
8.	How does the performance of Voxel Mamba degrade with respect to various levels of occlusion or truncation of objects in the point cloud data?

**Limitations:**

1.	The paper does not mention the issue of computing resource consumption, or whether higher computing resources are required.
2.	The paper does not explicitly discuss the model's generalization ability, that is, how well the model performs in new scenarios or under different environmental conditions. The paper's focus on autonomous driving scenarios may limit the general applicability of the findings to other fields where 3D object detection is relevant.
3.	The reliance on large-scale datasets for training and evaluation might not fully capture the diversity of real-world scenarios, potentially limiting the model's robustness.
4.	The potential impact of adversarial attacks or model overfitting to the specific characteristics of the training datasets is not discussed, which is important for model security and reliability.
5.	The performance of Voxel Mamba on the Waymo Open and nuScenes datasets may not directly translate to other types of datasets, potentially limiting the model's universal applicability.
6.	It is not clear whether the current Voxel Mamba model can be extended to a multi-task learning framework, such as simultaneously performing object detection, classification, and pose estimation tasks.

---

> ### Author Rebuttal · Authors · 2024-08-07
>
> **[Q1] Clarifying our motivation.**
>
> Thanks for the question. Please kindly refer to our responses to **Q1 of Reviewer kB9g** for details.
>
> **[Q2] Efficiency and performance for different model sizes and resolutions.**
>
> We appreciate the reviewer's feedback and conducted additional experiments. The results regarding the model size (i.e. feature dimension) and BEV resolution (voxel size) are shown in the following two tables.
>
> |Model Size|L2 mAPH|Vehicle(L1/L2)|Pedestrian(L1/L2)|Cyclist(L1/L2)|Latency(ms)|Memory(GB)|
> |----------|-------|--------------|------------------|--------------|-----------|----------|
> |dim=96|71.3|78.7/70.3|83.6/76.2|76.3/73.6|84|3.6|
> |dim=128|71.6|79.0/70.7|84.0/76.7|76.5/73.7|90|3.7|
> |dim=192|72.4|79.2/70.8|84.7/77.5|77.8/74.9|102|3.9|
>
> The above table demonstrates a consistent performance improvement as the dimension increases, which correlates with the model scale.
>
> |Voxel Size|L2 mAPH|Vehicle(L1/L2)|Pedestrian(L1/L2)|Cyclist(L1/L2)|Latency(ms)|Memory(GB)|
> |----------|-------|--------------|------------------|--------------|-----------|----------|
> |0.24m|71.5|78.7/70.4|84.0/76.3|76.9/74.0|114|5.4|
> |0.32m|71.6|79.0/70.7|84.0/76.7|76.5/73.7|90|3.7|
> |0.40m|71.4|79.0/70.7|83.6/76.2|76.5/73.8|81|3.2|
>
> The above table demonstrates that Voxel Mamba remains effective across different voxel sizes. Increasing voxel size reduces input BEV resolution, enhancing efficiency. Voxel Mamba performed best with a voxel size of 0.32m.
>
> **[Q4] Generalization of Voxel Mamba.**
>
> Thanks for your question. Please kindly refer to the **shared response Q3**.
>
> **[Q5] More comparison with other state-of-the-art methods.**
>
> We would like to draw attention to Figure 3 in our manuscript, which provides a comprehensive comparison of our method with state-of-the-art approaches commonly benchmarked in the field of 3D object detection.
>
> **[Q6] Robustness to density and sparsity.**
>
> We re-benchmarked the performance in different range intervals on Waymo to assess Voxel Mamba's robustness to density and sparsity. The following table indicates that Voxel Mamba is more robust than DSVT.
>
> | Method         | Overall | [0, 30m) | [30m, 50m) | [50m, inf) |
> |----------------|---------|----------|------------|------------|
> | DSVT           | 69.69   | 83.80    | 67.73      | 49.38      |
> | Voxel Mamba    | 71.61   | 85.51    | 70.31      | 50.60      |
>
> **[Q7] Motivation of downsampling in ASSMs.**
>
> Though space-filling curves can preserve the 3D structure to a certain degree, proximity loss is inevitable due to the dimension collapse from 3D to 1D. As a result, a local snippet of the curve can only cover a partial region in 3D space. Placing all voxels in a single group cannot ensure that the effective receptive field (ERF) could cover all voxels. Therefore, we introduce downsampling (hierarchy of state space structures) and consequently improve the ERF of the model.
>
> **[Q8] Clarification on pipeline.**
>
> Please kindly refer to our responses to Q4 of Reviewer kB9g for details.
>
> **[Q9] Clarification on title.**
>
> We respectfully disagree with your opinion. To clarify, while 3D object detection is often associated with point clouds, it is not exclusively a point cloud-based task.
>
> **[Q10] Computational complexity with point densities.**
>
> Refer to **shared responses** for details. With increased point densities (4-frame), Voxel Mamba is 20ms faster than DSVT, which is more pronounced in 1-frame and indicates our method's efficiency.
>
> **[Q11] Advantage over Transformers and Spconv.**
>
> For details on Transformers, please kindly see **Q1 of Reviewer kB9g**. It is important to note that traditional CNNs are typically avoided in 3D detection due to slow inference on sparse data. Compared to SparseConv, Voxel Mamba has significantly larger ERFs, enabling better capture of long-range dependencies in point clouds. Additionally, our method is more deployment-friendly on autonomous driving than previous methods since no padding tokens are needed.
>
> **[Q12] Motivation of IWPE.**
> The window-based position embedding has been proven effective in previous window-based transformers, as it retains the proximity of voxels within a local window. As a group-free model, we do not explicitly partition the voxels into windows. Instead, we implicitly encode their positions within the window as additional information in the voxel sequence, which significantly enhances the SSM model's ability to extract useful information from the context.
>
> **[Q13] Dynamic object.**
>
> As indicated in Table 1 of our manuscript, all our experiments were conducted in a single-frame setting. Consequently, the inputs do not include dynamic objects, as each frame is processed independently. To address the concern about non-static elements, we report the multi-frame results in the **shared response**.
>
> **[Q14] Degrade with truncation.**
> To evaluate Voxel Mamba's robustness to truncation, we compared it with DSVT under various truncation levels using the cutout methods in [1]. The table below shows the mAP of Voxel Mamba and DSVT on the nuScene dataset with truncation severities ranging from 1 to 5. One can see that Voxel Mamba consistently outperforms DSVT across all truncation densities, indicating our method's superior robustness.
>
> | Method        | Clean | Severity 1 | Severity 2 | Severity 3 | Severity 4 | Severity 5 |
> |---------------|-------|------------|------------|------------|------------|------------|
> | DSVT          | 66.4  | 65.1       | 64.7       | 63.1       | 62.2       | 60.5       |
> | Voxel Mamba   | 67.5 (+1.1) | 66.4 (+1.3) | 65.8 (+1.1) | 64.9 (+1.8) | 63.5 (+1.3) | 61.8 (+1.3) |
>
> [1] Benchmarking Robustness of 3D Object Detection to Common Corruptions in Autonomous Driving. CVPR23
>
> **[Q15] Adversarial attacks, dataset overfitting, and multi-task learning**
>
> Thanks for your suggestion. In this paper, we aim to establish a strong backbone in 3D object detection, and those issues will be discussed in future work.

---

> > ### Comment · Area_Chair_SwCv · 2024-08-11
> >
> > Dear Reviewer,
> >
> > This is a gentle reminder to please review the rebuttal provided by the authors. Your feedback is crucial to the decision-making process. Please consider updating your score after reading the rebuttal.
> >
> > Thank you for your help with the NeurIPS!
> >
> > Best, Your AC

---

> ### Author Response · Authors · 2024-08-11
> **We are open to any further discussion**
>
> Dear reviewer:
>
> Thank you so much for reviewing our paper. We hope our explanation could resolve your concern. During this discussion phase, we welcome any further comments or questions regarding our response and main paper. If there requires further clarification, please do not hesitate to bring them up. We will promptly address and resolve your inquiries. We are looking forward to your feedback.

---

> ### Comment · Reviewer_vX3v · 2024-08-13
>
> The author's rebuttal addressed many of the issues I raised. However, I still have some doubts about this article. Although the author has further elaborated on the motivation, there are still some areas of doubt. For visual perception tasks such as object detection, is contextual information really important, and what is the interpretability of using RNN type network frameworks. Overall, the paper is quite innovative, so I'm inclined to rate it as boardline acceptable.

---

> ### Author Response · Authors · 2024-08-14
>
> We really appreciate for this reviewer's recognition on the innovation of our work.
> Yes, the contextual information is important in point cloud object detection. Since 3D point clouds lack distinctive textures and appearances, the detectors will more heavily rely on contextual information for learning semantic information. We have developed several designs to enhance Voxel Mamba's ability to capture contextual information in the paper. To increase contextual information for voxel tokens, we proposed a group-free strategy to serialize the whole space voxels into a single sequence. Besides, we also integrated a multi-scale design to broaden the effective receptive field and capture context information at various scales.
>
> We used RNN to address the inefficiency of Transformers when expanding input context. In the large context setting, we adopted Mamba (RNN) design to enhance the efficiency, as latency is crucial in outdoor 3D object detection for autonomous driving. As shown in the table following, RNNs have a significant speed advantage over Transformers with the longer input voxel sequences, which often reach the order of $10^5$ in 3D detection scenarios.
>
> | Method     | 1K       | 2K       | 4K       | 8K        | 16K        |
> |------------|----------|----------|----------|-----------|------------|
> | Transformer| 0.47 ms  | 1.61 ms  | 5.74 ms  | 26.02 ms  | 114.20 ms  |
> | SSM        | 0.41 ms  | 0.43 ms  | 0.50 ms  | 0.61 ms   | 0.94 ms    |
>
> If there is any misinterpretation of your question, we are open to further discussion

---

### Official Review · Reviewer_2cxV · 2024-07-09

**Soundness:** 3
**Presentation:** 3
**Contribution:** 3
**Rating:** 7
**Confidence:** 5

**Summary:**

This paper proposes a novel backbone named Voxel Mamba for point cloud 3D detection. Different from previous methods that group the voxels into fixed-length sequences through padding, this paper adopt an interesting group-free strategy to sort all voxels into a single sequence through the space-filling curve. The paper also proposes the Asymmetrical State Space Models (ASSMs) and Implicit Window-based Position Embedding (IWPE) to enlarge the receptive fields of Voxel SSMs.
Experiments are mostly based on challenging outdoor datasets, the Waymo Open Dataset and nuScene. Experiments demonstrates Voxel Mamba surpasses previous methods.

**Strengths:**

1.	The paper presents a novel approach to deal with irregularity problem of point cloud detection. By taking the advantage of linear complexity of SSM, this paper employ a group-free strategy which is more efficient and deployment-friendly than previous group-based methods.
2.	ASSMs and IWPE are well motivated. The multi-scale design and implicit window partition are integrated under group-free design in an economical way. Besides, the figures in this paper are clear and greatly aid in the comprehension of the proposed methods.
3.	The proposed Voxel Mamba exhibits a notably larger ERF than group-based methods DSVT.  The visualization effectively supports their claim.
4.	The experiments are thorough and convincingly demonstrate the effectiveness of the proposed method.

**Weaknesses:**

1.	The experiments of model architectures are a bit insufficient. In this paper, the authors extend Mamba to bidirectional Mamba (line 68) and further to Voxel Mamba (line 74). However, there are no experiments demonstrating the performance gains from these upgrades. As Voxel Mamba is based on Mamba, it would be necessary to report the accuracy of apply group-free bidirectional Mamba and Mamba with the same SSM parameters or layers.
2.	Given that the group-free strategy is a significant contribution of this paper, it is necessary to include more analysis and comparisons between group-free and group-based operations. For example, the author cloud provide a detailed latency comparison between window-based grouping (such as dsvt and sst) and HIL. Besides, on Line 188, the authors mention that the mapping processes takes 0.7ms. However, the latency is likely to vary with different sequence lengths. Please provide a clarification regarding the timing.
3.	Lack of comparison with non-group-free Mamba. Although this work provides quantitative and visualization comparison between group-free and group-based (dsvt) methods. It is necessary to provide the ERFs and detection accuracy for non-group-free Voxel Mamba. As a linear RNN models, what difference does Mamba introduce compared to Transformers.
4.	The analysis of the ablation on ASSM and HIL is a little vague. Specifically, it is unclear why performances drop under the downsampling setting {2,2,2} and {4,4,4} in Table 5(b). Besides, the window sweep, Z-order, and Hilbert methods in Table 5(d) exhibit similar performance. Can I infer that the form of the space-filling curve does not significantly affect the accuracy? Please provide a discussion on them, which would be helpful to improve the quality of this paper.
5.	Hilbert Input Layer (HIL) sort the voxels by querying the space-filling curve template. As presented in Sec. 3.3, form my understanding, the template needs cover the whole scenes. My concern is the amount of GPU memory required by the template under the current voxel size setting and its contribution to the total memory consumption. And for a more fine-grained size such as (0.1m, 0.1m, 0.15m) in CenterPoint.

**Questions:**

Please see Weaknesses above. Overall, the proposed group-free strategy has better efficiency than previous group-based operations without token padding. Voxel Mamba also demonstrates better performance than existing sparse convolution or transformer structures. As such, I recommend acceptance at this stage.

**Limitations:**

Limitations have been included

---

> ### Author Rebuttal · Authors · 2024-08-07
>
> **[Q1] Comparison with other variants.**
>
> Thanks for your insightful comments. To demonstrate the improvements of our method, we have conducted additional experiments comparing Voxel Mamba with group-based and group-free bidirectional Mamba. The group-based bidirectional Mamba uses the DSVT Input Layer to partition the voxel groups. The group-free bidirectional Mamba is based on our Hilbert Input Layer. Each variant retains the same number of SSM layers for consistency in comparison.
>
> | Method | L2 mAPH | Vehicle (L1) | Vehicle (L2) | Pedestrian (L1) | Pedestrian (L2) | Cyclist (L1) | Cyclist (L2) |
> |:--|--:|--:|--:|--:|--:|--:|--:|
> | Group-based bidirectional Mamba | 68.5 | 76.1 / 75.7 | 67.8 / 67.4 | 81.8 / 75.8 | 74.2 / 68.6 | 73.3 / 72.3 | 70.6 / 69.6 |
> | Group-free bidirectional Mamba | 71.0 | 78.3 / 77.9 | 70.0 / 69.6 | 83.0 / 78.1 | 75.6 / 70.9 | 76.3 / 75.2 | 73.4 / 72.4 |
> | Voxel Mamba | 71.6 | 79.0 / 78.5 | 70.7 / 70.2 | 84.0 / 79.1 | 76.7 / 72.0 | 76.5 / 75.4 | 73.7 / 72.7 |
>
> The above table shows that enhanced spatial proximity can significantly improve the detector's performance. Additionally, our innovative multi-scale design yields additional performance gains, which can be attributed to the expansion of ERFs.
>
> **[Q2] Advantage over group-based operations.**
>
> Thanks for your suggestions. Please kindly refer to **shared response Q1** for the advantage of group-free operations.
>
> **[Q3] Comparison with group-based Mamba.**
>
> To address the reviewer's concern, we compared the performances between group-free and group-based mamba (refer to the table in our response to Q1). The results demonstrate the effectiveness of our group-free strategy, which can enhance the 3D proximity in the 1D sequences. Besides, we provide the ERFs of group-based (DSVT partition) Mamba in Figure 1 in the rebuttal PDF.
>
> **[Q4] Analysis on ASSMs.**
>
> We appreciate the reviewer’s insightful comment. The transitioning from \{1,1,1\} to \{1,2,2\} and to \{1,2,4\} enhances performance due to an enlarged effective receptive field and improved proximity by using larger downsampling rates at late stages. ASSMs with \{2,2,2\} or \{4,4,4\} compromise performance compared to \{1,1,1\}, indicating that using larger downsampling rates at early stages will lose some fine details. Thus, we set the stride as \{1,2,4\} to strike a balance between effective receptive fields and detail preservation. Additionally, your inference about the similar performance of different space-filling curves is astute. Voxel Mamba is designed with significantly large ERFs, which indeed diminishes the impact of the type of space-filling curves on performance. We will expand our analysis in revision.
>
> **[Q5] Curve template memory.**
>
> Yes, we record the traversal position in the space-filling curves of all potential voxels offline. The curve templates are adaptively generated based on the input BEV resolution. To address the reviewer's concern, the table below lists GPU memory usage for templates. The input BEV resolution is (468, 468), with three scale curve templates for each ASSM scale.
>
> | Template Resolution | Memory (MB) |
> |---------------------|-------------|
> | (512, 512)          | 83.0        |
> | (256, 256)          | 8.6         |
> | (128, 128)          | 1.2         |

---

> > ### Comment · Area_Chair_SwCv · 2024-08-11
> >
> > Dear Reviewer,
> >
> > This is a gentle reminder to please review the rebuttal provided by the authors. Your feedback is crucial to the decision-making process. Please consider updating your score after reading the rebuttal.
> >
> > Thank you for your help with the NeurIPS!
> >
> > Best, Your AC

---

> > > ### Comment · Reviewer_2cxV · 2024-08-12
> > >
> > > Thanks for the author’s comprehensive response. These address my concerns. The additional experiments show the advantages of group-free strategy and the effectiveness of ASSMs. The curve templates do not significantly impact GPU memory consumption. This work presents a meaningful advancement in 3D detection.

---

> > > > ### Author Response · Authors · 2024-08-13
> > > >
> > > > We sincerely thank this reviewer for the feedback and support. Your constructive comments are valuable for us to improve the quality of this work. Following your suggestions, we will polish the paper and add necessary clarifications in the revision.

---

### Official Review · Reviewer_kB9g · 2024-07-16

**Soundness:** 3
**Presentation:** 3
**Contribution:** 2
**Rating:** 7
**Confidence:** 5

**Summary:**

The authors introduce Voxel Mamba, a 3D object detection method based on state-space models.  Voxel Mamba introduces a efficient group-free approach to avoid inefficient grouping operations and prevent limitations on the receptive field for 3D object detection in point clouds. The authors also present Asymmetrical State Space Models (ASSMs) to capture multi-scale context information, offering each voxel data-dependent global contexts. Voxel Mamba enables voxel interactions throughout the scene and captures cloud locality using space-filling curves. Extensive experiments validate its performance, positioning Voxel Mamba as a viable alternative for 3D object detection.

**Strengths:**

1. The paper is well-organized and easy to follow.
2. The authors first time delve into the potential of Mamba in 3D object detection, which is an intriguing subject.

**Weaknesses:**

1. Motivation can be improved. The authors must clarify why SSM is essential for their motivation and quantify the advantages it offers over transformers for this task. The question in lines 55-56 does not demonstrate SSM's superiority. The author's design for SSM is also hardly inspiring for the 3D detection community.
2. Some of the method's design confuse me. Traversing the downsampled feature maps with a Hilbert curve in the opposite direction can misalign the feature receptive fields. Does this challenge model training with varying inputs, even with IWPE included? Additionally, given its use of multi-scale features, how much does ASSM outperform FPN?
3. The experimental comparisons may be inadequate. The authors present a pillar version for Voxel Mamba but fail to compare its performance and efficiency against current state-of-the-art pillar-based methods in the main table.
4. The experiment's details should be more clearly documented. For example, does Voxel Mamba adopts IoU-rectification scheme? The comparative setup in the ablation study could have been accounted for in more detail and analyzed why the method designed by the authors was effective.
5. How does Voxel Mamba integrate with multi-frame temporal 3D detection?

**Questions:**

See Weakness. Besides, there are few other suggestions:

1. Could the authors provide more description of Window Sweep in Table 5(c)? Does it include the Shifted Regional Grouping Shifted Token Groups operation used in SST?
1. Is a Linear for the up dimension missing from the SSM part of Fig. 2?

**Limitations:**

The authors have discussed the limitations.

---

> ### Author Rebuttal · Authors · 2024-08-07
>
> **[Q1] Motivation.**
>
> Thanks for the question and we are sorry for not making the motivation clear enough.  The motivation of our work is to leverage the advantages of SSM in linear attention for more effective feature learning in 3D point cloud based object detection.  The advantage of SSM over Transformer lies in its ability to more efficiently explore large context (as shown in Fig. 4 of the main paper), i.e., an extremely long voxel sequence, for effective feature learning. From the table below, one can see that for more than 4K voxel contexts, a single transformer layer consumes more than 5ms, which is impractical for perception tasks. However, even for a sequence of 16K voxels, SSM only consumes 0.94ms.
>
> | Method       | 1K     | 2K     | 4K      | 8K      | 16K     |
> |--------------|--------|--------|---------|---------|---------|
> | Transformer  | 0.47 ms| 1.61 ms| 5.74 ms | 26.02 ms| 114.20 ms|
> | SSM          | 0.41 ms| 0.43 ms| 0.50 ms | 0.61 ms | 0.94 ms  |
>
> Unlike previous window-based voxel transformers, we are the first to model the voxel sequence from the entire scene. By utilizing Mamba's selective scan mechanism, we can effectively retain useful information from the context without the need for complex voxel grouping. This group-free approach is insightful for handling large-scale sparse data.
>
> **[Q2] Effectiveness of multi-scale design.**
>
> We appreciate the reviewer's insightful observation. In fact, as a sequence-based model, the receptive fields of voxel features only rely on the sequence length, and they do not necessarily need to be aligned. Mamba can selectively retain useful information after scanning the sequence back and forth. Here we further compare ASSM with FPN on the Waymo dataset with 20% training data. Following Voxel Mamba, we utilize Spconv and its counterpart SpInverseConv as downsampling and upsampling operations. We keep the same downstrides in both structures. The table below shows that our ASSM achieves superior performance in the 3D object detection tasks.
>
> | Model Size            | L2 mAPH | Vehicle (L1) | Vehicle (L2) | Pedestrian (L1) | Pedestrian (L2) | Cyclist (L1) | Cyclist (L2) |
> |-----------------------|---------|--------------|--------------|-----------------|-----------------|--------------|--------------|
> | Mamba with FPN        | 68.2    | 77.4 / 76.9  | 69.1 / 68.6  | 81.3 / 74.0     | 73.8 / 66.9     | 72.1 / 71.0  | 69.4 / 68.3  |
> | Vxoel-Mamba-Pillar    | 69.5    | 78.1 / 77.6  | 69.8 / 69.3  | 82.4 / 75.5     | 75.0 / 68.5     | 74.6 / 73.4  | 71.8 / 70.7  |
>
> **[Q3] Comparison with state-of-the-art pillar-based methods.**
>
> Thanks for the suggestion. The following table compares our pillar-based Voxel Mamba with previous state-of-the-art pillar-based methods. We can see that Voxel Mamba outperforms previous methods on both L2 mAP and mAPH while achieving comparable inference speed.
>
> | Method                 | L2 mAP/mAPH | Vehicle (L1) | Vehicle (L2) | Pedestrian (L1) | Pedestrian (L2) | Cyclist (L1) | Cyclist (L2) | Latency (ms) |
> |------------------------|-------------|--------------|--------------|-----------------|-----------------|--------------|--------------|--------------|
> | SST                    | -           | 76.2 / 75.8  | 68.0 / 67.6  | 81.4 / 74.1     | 72.8 / 65.9     | -            | -            | 71           |
> | DSVT-Pillar            | 73.2 / 71.0 | 79.3 / 78.8  | 70.9 / 70.5  | 82.8 / 77.0     | 75.2 / 69.8     | 76.4 / 75.4  | 73.6 / 72.7  | 64           |
> | Voxel Mamba-Pillar     | 73.5 / 71.2 | 80.2 / 79.7  | 72.1 / 71.7  | 83.5 / 77.6     | 76.1 / 70.5     | 75.0 / 74.0  | 72.2 / 71.3  | 66           |
>
> **[Q4] Detailed experiments and ablation studies.**
>
> We apologize for any confusion regarding the details of our experiments. For fair comparison, our experiment settings follow the schemes in DSVT, using the Adam optimizer with weight decay of 0.05, a one-cycle learning rate policy, and a maximum learning rate of 2.5e-3. Models were trained with a batch size of 24 for 24 epochs on 8 NVIDIA A6000 GPUs. During inference, we use class-specific NMS with IoU thresholds of 0.7, 0.6, and 0.55 for vehicles, pedestrians, and cyclists, respectively. Ground-truth copy-paste data augmentation was used during training and disabled in the last epoch. Our setting aligns closely with the previous state-of-the-art DSVT, and the primary difference is on the backbone. In addition, **Voxel Mamba does not adopt the IoU-rectification scheme**.
>
> In our ablation study, we assessed the effectiveness of various components and drew the following conclusions:
> (a) Space-filling curves enhance spatial proximity and locality, significantly improving performance (Table 5d).
> (b) Position embedding with IWPE boosts detection performance by providing rich 3D positional and proximate information (Table 5a).
> (c) Downsampling rates of ASSM show that larger effective receptive fields improve performance, but very large downsampling rates at early stages can degrade performance. We found a balance with a stride of {1,2,4} (Table 5).
> (d) Bidirectional SSMs with a Hilbert-based group-free sequence significantly improves accuracy, validating our group-free strategy. ASSM enhances effective receptive fields and proximity, while IWPE further boosts Voxel Mamba’s performance by capturing 3D positional information (Table 6c).
>
> **[Q5] Multi-frame results.**
>
> Please refer to the **shared responses Q2** for details.
>
> **[Q5] Description of window sweep.**
>
> The window sweep adopts the same regional grouping method as SST, extended to 3D voxels.
> We use a fixed BEV window size of (12, 12) and implement Region Shift between each two ASSM blocks.
>
> **[Q6] Detail of Figure 2.**
>
> We apologize for this problem. We will revise Figure 2 to accurately reflect the complete architecture.

---

> > ### Comment · Area_Chair_SwCv · 2024-08-11
> >
> > Dear Reviewer,
> >
> > This is a gentle reminder to please review the rebuttal provided by the authors. Your feedback is crucial to the decision-making process. Please consider updating your score after reading the rebuttal.
> >
> > Thank you for your help with the NeurIPS!
> >
> > Best, Your AC

---

> ### Author Response · Authors · 2024-08-11
> **We are open to any further discussion.**
>
> Dear reviewer:
>
> Thank you so much for reviewing our paper. We hope our explanation could resolve your concern. During this discussion phase, we welcome any further comments or questions regarding our response and main paper. If there requires further clarification, please do not hesitate to bring them up. We will promptly address and resolve your inquiries. We are looking forward to your feedback.

---

> > ### Comment · Reviewer_kB9g · 2024-08-12
> >
> > Thanks for your careful responses and clarification, which have effectively addressed my concerns and misunderstandings. I have read the authors' responses and other reviewers' comments.
> >
> > It is good to see that the unclear experiments' detail and some ablation study analysis have been supplemented. The additional experiment and comparison, which demonstrate the efficiency of Voxel Mamba's structure and SSM, have been provided during the rebuttal.
> >
> > I strongly encourage the author to incorporate the additional experiments, especially the comparison between Voxel Mamba and DSVT input layer, from the rebuttal into the revised manuscript.
> >
> > Given these improvements and clarification, I have an overall favorable opinion and upgrade my rating.

---

> > > ### Author Response · Authors · 2024-08-13
> > >
> > > We sincerely thank this reviewer for finding our responses useful and raising the score. We will incorporate additional experiments and detail explanations in the revised manuscript, as suggested by this reviewer. Thanks again for your constructive comments!

---

### Official Review · Reviewer_dTn9 · 2024-07-25

**Soundness:** 3
**Presentation:** 4
**Contribution:** 3
**Rating:** 7
**Confidence:** 4

**Summary:**

This paper proposes a new 3D detection architecture, Voxel Mamba. Voxel Mamba serializes voxels using a Hilbert Input Layer and then applies a forward SSM and a backward SSM at lower resolutions. It achieves state-of-the-art performance on both the Waymo Open Dataset and nuScenes dataset.

**Strengths:**

Originality:

Addresses the challenge of long sequence lengths in group-free 3D detection models by using the Mamba SSM module to extract features from all voxels, avoiding inefficient sorting and grouping operations found in window-based sparse 3D detectors.

Quality:

 Demonstrates strong 3D detection performance on both Waymo Open Dataset and nuScenes, supported by extensive ablation studies in Table 5, highlighting the effectiveness of Voxel Mamba's design choices.

Clarity:

The paper is well-written with clear visualizations, notably Figure 4, which illustrates the larger effective receptive field of Voxel Mamba compared to group-based methods.

**Weaknesses:**

Please see the questions section.

**Questions:**

In table 5 (d), the Hilbert curve is marginally better than the Z-order curve. In 5(c), adding Hilbert improves mAP and NDS by 0.1.
Curious, what is the latency of generating the Hilbert curve and Z-order Curve. What about the latency of generating Random Curves?

For random curves, do you use the same random curves for all frames or random sample a curve for each frame? Do you use the same random sampled curves for different resolutions, or use the same random curve for all resolutions in the backward SSM?

What would be the expected performance and latency if you flattened the voxels to 1D but used window attention (either shifted or scanned) instead of Mamba? This could help assess the trade-offs between different approaches to handling long sequence lengths.

**Limitations:**

The authors adequately addressed the limitations.

---

> ### Author Rebuttal · Authors · 2024-08-07
>
> **[Q1] Latency of generating curves.**
>
> In our implementation, we record the traversal position in the space-filling curves of all potential voxels offline. The voxels are serialized by simply looking up these traversal positions. Thus, the latency for serializing voxels based on Hilbert and Z-order curves is identical. The detailed generation times of the Hilbert Input Layer are shown in the **shared responses Q1**.
>
> **[Q2] Random Curves setting.**
>
> In the Hilbert Input Layer (HIL) implementation,  we use fixed templates for each resolution in the ASSMs. With the downsampling rate {1,2,4} in ASSMs, Voxel Mamba requires three different scale curve templates.
>
> However, the "random" in Table 5(a) means that we do not perform any sorting on the input sequence for SSMs. We observed that the Voxel Feature Extractor (VFE) and the downsampling and upsampling operations tend to preserve the spatial proximity of neighboring voxels in the 3D space in the 1D sequence. This inherent characteristic leads to a favorable outcome for SSMs. To ensure a real random order, we generate three random templates and use them for all frames following the HIL setting.
> The following shows the results on the Waymo open dataset with 20% training data. These results indicate that disrupting voxel proximity significantly degrades performance, highlighting the effectiveness of our Hilbert Input Layer.
>
> | Method       | L2 mAPH | Vehicle (L1) | Vehicle (L2) | Pedestrian (L1) | Pedestrian (L2) | Cyclist (L1) | Cyclist (L2) |
> |--------------|---------|--------------|--------------|-----------------|-----------------|--------------|--------------|
> | Random order | 51.7    | 56.4 / 55.7  | 49.1 / 48.4  | 67.0 / 57.9     | 58.6 / 50.6     | 60.4 / 58.2  | 58.1 / 56.0  |
> | Hilbert      | 71.6    | 79.0 / 78.5  | 70.7 / 70.2  | 84.0 / 79.1     | 76.7 / 72.0     | 76.5 / 75.4  | 73.7 / 72.7  |
>
> **[Q3] Hilbert Input Layer + attention.**
>
> We truly appreciate this valuable comment. We conducted an experiment by replacing the window grouping in DSVT with the serialization approach in Voxel Mamba. We use the same group size of 48 as DSVT, and only pad the last voxel groups for each sample to enable the parallel computation of Transformers. We employ curve shift as an alternative to the X/Y-axis partition and perform experiments on 20% Waymo training data. The following table compares this variation with Voxel Mamba and DSVT.
>
> | Method                   | L2 mAPH | Vehicle (L1) | Vehicle (L2) | Pedestrian (L1) | Pedestrian (L2) | Cyclist (L1) | Cyclist (L2) | Latency (ms) |
> |--------------------------|---------|--------------|--------------|-----------------|-----------------|--------------|--------------|--------------|
> | DSVT                     | 69.7    | 77.9 / 77.4  | 69.5 / 69.1  | 82.2 / 76.3     | 74.8 / 69.1     | 74.7 / 73.6  | 71.9 / 70.9  | 94           |
> | Hilbert + self-attention | 68.9    | 77.3 / 76.8  | 69.0 / 68.5  | 82.0 / 75.7     | 74.5 / 68.7     | 73.4 / 72.3  | 70.7 / 69.6  | 85            |
> | Voxel Mamba              | 71.6    | 79.0 / 78.5  | 70.7 / 70.2  | 84.0 / 79.1     | 76.7 / 72.0     | 76.5 / 75.4  | 73.7 / 72.7  | 90           |
>
> One can see that this variant with the Hilbert Input Layer shows slightly lower performance than DSVT. We hypothesize that this drop is due to the limited group size in the window attention mechanism. This constraint might limit the full potential of the HIL voxel flattening. While it does not outperform Voxel Mamba in accuracy, it does offer a favorable trade-off between performance and latency.

---

> > ### Comment · Area_Chair_SwCv · 2024-08-11
> >
> > Dear Reviewer,
> >
> > This is a gentle reminder to please review the rebuttal provided by the authors. Your feedback is crucial to the decision-making process. Please consider updating your score after reading the rebuttal.
> >
> > Thank you for your help with the NeurIPS!
> >
> > Best, Your AC

---

> > ### Comment · Reviewer_dTn9 · 2024-08-12
> >
> > Thank you for providing additional results. I will keep my rating.

---

> > > ### Author Response · Authors · 2024-08-13
> > >
> > > Thank you very much for your dedicated efforts to review our paper and suggestions for our work.

---

### Author Rebuttal · Authors · 2024-08-07

We sincerely thank all reviewers for the valuable comments and suggestions. We first address the common concerns, followed by the detailed responses to each reviewer separately. We hope our responses can clarify the reviewers' concerns and make our contributions clearer.

**Q1. The latency comparison between group-free and group-based operations**

We perform a detailed comparison of the Hilbert Input Layer (HIL) and DSVT Input Layer across different point cloud sizes and densities. To ensure that the distributions of LiDAR sensor data can be well described, we stack frames to gradually increase the point cloud density. The following table shows the runtime of serialization with the same voxel size (0.32m, 0.32m, 0.1875m) on the Waymo open dataset. One can see that compared to window-based grouping operations, our group-free approach is 13.74ms faster in single-frame scenarios. Moreover, the advantage of HIL becomes more pronounced with the increase of point cloud density.

| Method | 1-frames | 2-frames | 3-frames | 4-frames |
|--------|----------|----------|----------|----------|
| X/Y-Axis Grouping in DSVT | 19.41 ms | 32.30 ms | 34.10 ms | 36.78 ms |
| HIL in Voxel Mamba | 5.67 ms | 7.74 ms | 8.20 ms | 8.52 ms |

**Q2. Multi-frame setting performance**

| Method | Data | Latency (ms)        | L2 mAPH | Vehicle (L1) | Vehicle (L2) | Pedestrian (L1) | Pedestrian (L2) | Cyclist (L1) | Cyclist (L2)  |
|--------|------|--------------------|-------------|------------------|---|---------------------|---|-------------------|---|
| DSVT-4frames | 20% | 158.6 | 76.1 / 74.8 | 80.8 / 80.3 | 72.8 / 72.4 | 85.1 / 82.3 | 78.3 / 75.6 | 79.8 / 78.9 | 77.3 / 76.4 |
| DSVT-4frames | 100% | 158.6 | 76.9 / 75.6 | 81.8 / 81.4 | 74.1 / 73.6 | 85.6 / 82.8 | 78.6 / 75.9 | 80.4 / 79.6 | 78.1 / 77.3 |
| **Voxel Mamba-4frames** | 20% | 138.7 | **77.2 / 75.9** | 81.9 / 81.5 | 74.0 / 73.6 | 86.7 / 84.0 | 80.0 / 77.4 | 80.0 / 79.0 | 77.5 / 76.6 |

The above table compares the performance of DSVT and our Voxel Mamba under the multi-frame setting. One can see that Voxel Mamba significantly outperforms DSVT, indicating its superior ability to leverage temporal and spatial information across frames. Notably, Voxel Mamba trained on 20\% training data surpasses DSVT trained on the 100\% data.

**Q3. Indoor Semantic Segmentation**

To further demonstrate the general applicability of Voxel Mamba, we extended our experiments to the indoor 3D semantic segmentation task. We adopt the widely used encoder-decoder structure, with both components consisting of ASSMs. Following previous works, we partition the encoder into five stages, with the number of blocks in each stage being {2, 2, 2, 6, 2}. Downsampling is performed between each stage to reduce spatial dimension progressively. The feature dimensions of the blocks across these five stages are {32, 64, 128, 256, 512}. The downsampling rates for ASSMs' backward branches in each encoder stage are {1, 1, 1, 2, 2}. Our method is implemented based on the open-source framework Pointcept. The table below shows the semantic segmentation performance on Scannet. Our method outperforms Swin3D by 90.8% mIoU. Please note that due to limited time, this is just preliminary results. We believe better results can be obtained with refined design and implementation.

| Method                   | Val mIoU (%) |
|--------------------------|--------------|
| Stratified Transformer   | 74.3         |
| OctFormer                | 75.7         |
| Swin3D                   | 76.3         |
| Voxel Mamba              | 77.1         |

---

### Decision · Program_Chairs · 2024-09-25

**Decision:**

Accept (spotlight)

**Comment:**

This paper introduces an interesting adaptation of the Mamba architecture for the 3D detection task. The accuracy of the target task is impressive, and all readers appreciate the authors' contribution. Minor concerns regarding computation overhead, GPU memory consumption associated with Hilbert input layer (HIL), technical details, contribution analysis on the group-free strategy, additional comparison, typos, and explanation of the key ideas were raised during the rebuttal phase. The authors provided solid feedback to relieve the concerns. As a result, all reviewers recommend paper acceptance.